# A Pilot Study to Incorporate Collaboration and Energy Competency into an Engineering Ethics Course

**Yi-Chu Hsu**

Department of Mechanical Engineering, Southern Taiwan University of Science and Technology,
1 Nan-Tai Street, Yung Kang District, Tainan City 71005, Taiwan; yichu@stust.edu.tw

**Abstract:** According to the OECD, The Organization for Economic Co-operation and Development, and other education policy experts all over the world, an urgent reform is needed to promote education innovation with "competencies" as the core. To investigate the feasibility to apply competency-oriented education, this pilot study surveyed the competencies of "collaboration" and "energy" and applied competency-oriented contents into an Engineering Ethics course in the Department of Mechanical Engineering. The literature reveals that collaboration includes three constructs: trust, communication, and coordination. These constructs were used to develop a questionnaire and to survey the collaboration competency of the research subjects. In addition, an energy perception survey for Taiwan was used to compare and analyze the energy competencies between the research subject and the general adults in Taiwan. Finally, some suggestions are proposed for competency implementation in future courses.

**Keywords:** competency; collaboration; communication; coordination; energy literacy

## 1. Introduction

In the present age of knowledge proliferation, the flooding of fake news, and online visibility, the last job for a teacher is to give students more knowledge. Excessive information has crowded out students, leaving them stunned and confused. Students in the 21st century need to understand the information they are provided, judge whether it is true or false, screen which information is important, and integrate large amounts of meaningful data. Therefore, education policy experts worldwide, such as the Organization for Economic Co-operation and Development (OECD), the National Education Association (NEA), and the Center for Curriculum Redesign (CCR), have expounded the need to reform and promote education innovation with "competencies" as the core. "Competencies" refers to the knowledge, ability (including skills), and attitude that a person should have in order to adapt to his or her current life and face future challenges. Competencies emphasize that learning should not be limited to subject knowledge and skills but should focus on a combination of learning and life and demonstrate the whole-person development of learners through practice [1–3].

In 2018, the NEA recommended the four Cs (Collaboration, Critical thinking, Communication, and Creativity) in its publication, "Preparing 21st Century Students for a Global Society [1]." There seem to be four competencies here; however, these competencies are inextricably linked in the global society of the 21st century. A "collaborative" team should be able to "communicate" well and be full of "creativity" and "critical thinking". The four Cs thus present four faces of the talents needed in the near future. To initiate this research, only collaboration was investigated in this pilot study, although other competencies, like communication, which are constructs of collaboration were also investigated.

Implementing competency courses in higher education is an important issue. Competency is one of the most important parameters for students' employability indicators and is a relatively cross-domain comprehensive ability. It is difficult to evaluate a single subject objectively. However,

a hidden competence of collaboration can assist the learning of different professional knowledge and skills [2–4]. This study as a result uses teamwork as a competency learning tool for engineering students' course activities.

Many scholars have emphasized the importance of collaboration among the 4Cs. For example, Surowiecki [5] described the "crowd wisdom" in the new economy, where groups are often smarter than their smartest individuals. Surowiecki emphasized that, through a group of different people, even smarter decisions and predictions than those produced by the most skilled decision-makers can be made [5]. Diversity brings multiple personal and cultural perspectives into collaboration, and collaborative efforts not only create more comprehensive results than individual efforts but also create knowledge for more people. As a result, students' collaborative work can generate more knowledge, making collaboration a key factor for success in today's global society.

The NEA believes that "collaboration" includes the following characteristics [3]:

1. The ability to work effectively and respectfully with different teams;
2. The ability to demonstrate flexibility and the willingness to make the necessary compromises to achieve common goals;
3. The ability to take shared responsibility and value the individual contributions of each group member.

Traditional higher education is primarily focused on personal knowledge and skill development. However, high-achieving students in school may not be very successful in society. It could result from that school training focuses on individuals, but most of society's activities are team activities. Therefore, teamwork is required to get work done in the workplace. The following related studies show the research aspects of collaboration.

1. The degree of coordination between various departments is very important for the success of product development [6].
2. A highly trusted society has more opportunities for innovation [7].
3. Collaboration capabilities, including the three aspects of trust, communication, and coordination, have a significant impact on the performance of collaborative product innovation [8].
4. Trust is a key success factor for team innovation [9].
5. Trust exists when a partner has confidence in the reliability and integrity of the other partner [10].
6. Predictability, dependability, and faith are the three main aspects of trust [11].
7. Anderson and Narus (1990) define communication as providing immediate, accurate, and sufficient information in a way that the other partner can understand [12].
8. Beckett-Camarata, et al. (1998) believe that communication skills can reduce the uncertainty of the collaboration process and ensure a close cooperative relationship [13].
9. Mohr and Spekman (1994) regarded coordination as the integration of an organization's members, activities, routines, and work assignments in order to achieve the overall goal of the organization [14].

Based on the aforementioned literature, the collaboration competency includes the three constructs of trust, communication, and coordination [8]. The collaboration competency can help students improve their professionalism and achieve good academic performance, but few disciplines are used as the basis for in-depth study and evaluation in higher education. In addition, competency is often considered an implicit indicator. As long as the knowledge and skills required by the learning standards in higher education are acquired, it is expected that the quality of the students will meet the expectations of the aforementioned competencies, without much further investigation.

To further realize the feasibility to apply competency into higher education, energy was also chosen to be a subject to investigate the competency status of college students. First of all, the content of energy as the competency is suitable for the candidates of the research subject, the students of

Mechanical Engineering. Energy is fundamental to develop the most professions of Mechanical Engineering and their future professional jobs will relate to energy more or less inevitably. The students are therefore quite familiar with this subject and easily catch energy-related issues. The second and most important cause to apply energy competency into the curriculum of Engineering Ethics is owing to the characteristics of ethics which is ambiguous and broad if the course does not have a specified subject. That is the reason why many Engineering Ethics courses emphasize case study which can make ambiguous ethical topics solid. Furthermore, the course needs a specified goal to practice ethical issue; otherwise, the lecture will become a dogma which cannot change the behavior of the students and disobey the purpose of the course. Meanwhile, competency-oriented education required the lecture materials are applicable in real life and therefore it fits the goal of the Engineering Ethics course. In other words, engineering ethical course originally should be competency oriented; otherwise, it cannot achieve the course purpose.

Moreover, energy is the driving force of national civilization, and, therefore, countries around the world attach great importance to education in energy technologies [15–17]. The energy issue is not just about science and technology. It is also linked to socioeconomic concepts, national security, national health, and the ecological environment. No single major factor can dismantle the overall system because the system interacts with many fields and requires multidisciplinary professional support to deal with complex issues [15]. Hence, literacy education is important [18]. Using the energy education development projects in the United States as an example, the "National Energy Education Development Program (NEED)" and "K-12 Energy Education Program (KEEP)" have been employed to promote energy as a literacy-based educational program [16].

The proposal "Energy Literacy: Essential Principles and Fundamental Concepts for Energy Education", announced by the US Department of Energy, defines energy literacy as an understanding of the nature and role of energy in the universe and in our lives [15]. Energy literacy is the ability to use this understanding to answer and solve problems. The purpose of energy education is to give people energy literacy, who can then:

1.  Think in terms of energy systems and track the flow of energy;
2.  Know how much energy is used, as well as why it is used and where it comes from;
3.  Assess the credibility of energy information;
4.  Communicate views on energy and energy use in a meaningful way;
5.  Make decisions about energy and energy use based on an understanding of the relevant impacts;
6.  Continues to learn about energy throughout his or her life.

Energy has always been a critical issue in Taiwan due to the dilemma of 98% of energy dependence on imports. However, the nuclear-free homeland policy of 2025 dramatically forces Taiwan's energy transition and has even made the energy issue more challenging. Furthermore, the democratic consciousness in Taiwan is increasing; therefore, it is difficult to complete the energy transition process solely with centralized administrative orders. It is thus necessary to strengthen public opinion as the basis for a smooth transition. In response, citizen participation in risk communications has become an important national movement for the energy transition [19,20]. All government units are doing their best to ensure energy is available to illuminate Taiwan. The Ministry of Science and Technology has conducted important and ongoing research in the energy fields of science, technology, society, and communication. Many research cases are being produced every year [21,22]. The Ministry of Education has also widely implemented energy-related practice in the education system. For example, the Office of Talent Cultivation for the integration and application of clean energy systems has established seven clean energy system practice bases in universities across Taiwan. These programs allow teachers and students of different levels, and the public, to personally experience the relevance and importance of renewable energy in life, society, and industrial development. These programs provide a wealth of online digital courses, energy competitions, and lectures to facilitate the mass education of energy literacy [23]. However, their main functions are not to optimize college course curricula. In addition,

although energy technology is traditional, its ever-changing progress always makes it a new issue for college courses. Its transdisciplinary characteristics (including politics, society, the economy, people's livelihoods, security, etc.) make it especially difficult to study extensively in a single knowledge-based university course.

According to the literature reviews, the current trend is to treat competencies as the core of future education [1]. The 4Cs have been well studied [2–14] and are recommended by the world education organization for the future generation. Meanwhile, energy is the driving force for human civilization and a complex subject that integrates many science, technology, and social fields as a candidate platform to train and evaluate the competencies of students [15]. Energy literacy has been applied to the educational systems of elementary, high school [16], and post-secondary education worldwide [17,18] and in Taiwan [19–23]. Although the government has heavily invested in university research funding, most of the research cases are aimed at promoting energy education for the general public and assisting in the implementation of energy transformations with the purpose of risk communication. Few studies have been made to investigate the feasibility to apply energy competency into one college course. To inspect the feasibility of incorporating collaboration and energy competency into a college course, this pilot study used one Engineering Ethics course to carry out the following goals:

Collaboration and energy competencies were chosen and surveyed for the third- and fourth-year students in the Department of Mechanical Engineering to investigate their competence status;

A collaboration questionnaire was formed and this competency between the students of the Engineering Ethics course and the general Mechanical Engineering students were compared and studied;

An energy questionnaire was surveyed and this competency between the students of the Engineering Ethics course and the public people of Taiwan were compared and studied;

Apply competency-oriented contents into the Engineering Ethics course and propose suggestions for their implementation in a future course.

According to the survey results, this study found that coordination is weakest, while trust is strongest for the research subjects, among the three constructs of collaboration competency. In addition, the energy perception of the research subjects was largely similar to that of the people in Taiwan, except that the subjects may have better knowledge about topics like the major types of energy resources in Taiwan but lack the willingness to increase electricity prices to support clean energy. Finally, some strategies were proposed to deal with the issues encountered during research execution, like the confusion of dual-topic progression and how to reduce invalid questionnaire samples.

## 2. Research Framework

To promote the professional development of engineers, experts in the United States [24] and globally [25,26] have pointed out the importance of moral education. To ensure that ethics is a core component of accredited engineering courses, in 2000, the American Board of Accreditation for Engineering and Technology (ABET) stated that college graduates should "understand professional and ethical responsibilities." With the amendment of ABET in 2017–2018, the code of ethics has been extended to include the different impacts of engineering solutions in global, economic, environmental, and social contexts, and the ability to make wise decisions in the field of engineering with ethical and professional responsibility [27]. In addition, ethical education based on cognition alone can easily be dogmatic. Students will solely consider the knowledge needed for a test without achieving the goal of changing people's minds and behaviors. This will violate the expectations of the general public regarding ethical courses and higher education.

Two competency-oriented topics, collaboration and energy, were applied to an Engineering Ethics course to achieve the course's expectations. The structure of this research framework is the following: Collaboration concept and workshop implementation → Collaboration questionnaire self-assessment → Engineering ethics case teaching by teamwork learning → Energy literacy local case teaching by teamwork learning → Data collection and analysis → Revising the course design.

*2.1. Research Subjects*

The research subjects were 59 students who took the compulsory course "Engineering Ethics and Society" at the Department of Mechanical Engineering in a private university of science and technology in Taiwan. There are more males than females in this course, with males accounting for 93.1% (100% were in the college of engineering). Most students were in their third and fourth year, with 91.3% and 8.6%, respectively. The majority were 20 years old, comprising 81.0% of the study population.

*2.2. Research Tools*

The data collection portion of this study includes two elements: qualitative and quantitative. The tools and operating weeks used in the research project are listed in Table 1. This pilot study mainly analyzed two questionnaires: The teamwork self-evaluation in Week 5 and the Taiwan Energy Perception Survey in Week 11.

**Table 1.** List of course evaluation tools and operation week.

| Item | Qualitative | Quantitative |
| --- | --- | --- |
| Collaboration | Homework: Two workshops (4 *) | Questionnaire: Collaboration self-evaluation (5) |
| Energy Literacy | Homework: Visiting the energy issue fields (14), Final written report—Justice for Energy Transformation (16), Final PPT Group Report (17) Questionnaire: Taiwan Energy Perception Survey (11) | Questionnaire: Taiwan Energy Perception Survey (11) Class activities: Energy transition justice (11, 15) Survey of Justice and Equality Values (15) |

* The bracket indicates the week of the activity

*2.3. Course Design*

At the beginning of the course, collaboration was the main topic and gradually moved to a classic case of Engineering Ethics (BP Deepwater Horizon), which is also a topic related to energy technologies and society. In the second half of the semester, the theme of STS (Science, Technology, and Society) continued, with two local energy issues in Taiwan, Taoyuan Algae Reef, and Longzaki Business Waste Landfill Sites. Finally, the students collected data and organized the information for a final report, as listed in Table 2.

**Table 2.** The course design over 18 weeks.

| Weeks | Course Content | Evaluation |
| --- | --- | --- |
| 1–4 Collaboration | "Collaboration" One Day Workshop * "Communication, Empathy" One Day Workshop * Workshop reflection and sharing | Homework_"Collaboration" Workshop Homework_Communication, Empathy Workshop Questionnaire_Collaboration self-assessment |
| 5–9 | Case Study of Engineering Ethics—BP Deepwater Horizon | |
| 10–14 Local Energy Issues | Energy Transition Justice in the Algae Reef of Taoyuan Social Risk Management in Ryazaki Ryuzaki Field Walk Course * | |
| 15–18 Report preparation | Energy, Sustainability, and Risk Communication_Data Consolidation Definition Issues in the Citizen Community for Society and Technology | Homework_Final written report Final oral report in the group |

* This denotes atypical classroom courses, such as one-day workshops on weekends or half-day courses on site.

# 3. Research Results and Discussion

The results of the teamwork and energy surveys conducted in accordance with the course are discussed here.

*3.1. Collaboration Questionnaire*

This study follows the previous literature review as mentioned in the Introduction. Communication, trust, and coordination are the three constructs of the questionnaire. One questionnaire was designed, and a preliminary study was published [28]. In essence, a collaboration questionnaire that meets the goals of the plan with its three constructs was formed by rearranging the following literature data.

1.    Scoring criteria for effective team operation [29];
2.    A 5C scale-teamwork ability questionnaire [30];
3.    An Index of team study papers [31].

The questionnaire is divided into three parts. The first part features the "intra-group coordination" scale; the second part uses the "intra-group communication" scale; and the third part uses the "intra-group trust "scale. The construct measurement adopts a Likert-type seven-point measurement scale (from complete disagreement (one point) to complete agreement (seven points)) as its quantitative basis. Statistical analysis was performed on the collected questionnaires, and statistical methods, such as narrative statistics and project analysis, were performed using the SPSS 25.0 statistical software as the analysis tool.

To compare the results for the research subjects in the Engineering Ethics course, a total of 244 questionnaires were sent to the students of the five classes in the Mechanical Engineering Department in a private university of science and technology in Taiwan. The collected samples were screened, and invalid questionnaires with the same scores were excluded. A total of 176 valid questionnaires was obtained. In terms of gender, more males were tested than females, with 93.2% of males; 100% of colleges were engineering colleges; the students were in their third or fourth year (respectively, 86.4% and 13.6%); and most subjects were 20 years old, at 76.9%.

Among the five classes issued questionnaires, one of the classes was the Engineering Ethics class (the research subject). There were 58 total questionnaires in the study. Screening samples and invalid questionnaires with the same scores were excluded, and a total of 36 valid questionnaires were obtained.

The value range of different degrees of performance is shown in Table 3 for interval values that are 7/5 = 1.4 on a seven-point scale.

**Table 3.** The performance of a seven-point scale.

| Performance | Value Range |
|---|---|
| Low | From 1 to less than 1.4 |
| Low to medium | From 1.4 to less than 2.8 |
| Moderate | From 2.8 to under 4.2 |
| Medium to high | From 4.2 to under 5.6 |
| High | 5.6 points or higher |

In terms of the scale of recognition of the Engineering Ethics class in the intra-group coordination (Table 4), the mean of Item 1 is the highest with a value of 5.33, which indicates that the research subjects attached the greatest recognition to each meeting starting on time, followed by Item 3 with a mean of 5.19. In addition, the means of Items 4, 9, and 8 were all higher than the construct average of 5.01. The 10 questions presented mid- to high-level performance, ranging from 4.2 to 5.6 points. Meanwhile, comparing the Engineering Ethics class based on an average of the five classes, 8 items out of 10 had a recognition level lower than the means of the five classes. The average value in the column of the mean increases between the two groups is −0.09, and the standard deviation is 0.12. Only Items 1 and 10 of the Engineering Ethics class were higher than the means of the five classes, and Items 5, 7, and 8 were significantly lower than the means of the five classes. Finally, the mean of the constructs of

the intra-group coordination shows the overall recognition, 5.01, to be slightly lower than the mean of the five classes, 5.10.

**Table 4.** Data for the construct of intra-group coordination.

| Items | Mean | Standard Deviation | Mean Increase * |
|---|---|---|---|
| 1. Group meetings always starting on time. | 5.33 (5.23) ** | 1.45 (1.29) | 0.1 |
| 2. Team members arrive late, depart early, or never attend (reversed question). | 3.31 (4.1) | 1.52 (1.63) | −0.79 |
| 3. The meeting has clear objectives. | 5.19 (5.33) | 1.1 (1.14) | −0.14 |
| 4. Everyone prepares well for the meeting, such as doing preliminary research or completing assignments. | 5.06 (5.19) | 1.18 (1.11) | −0.13 |
| 5. Everyone is interested in participating in group meetings. | 4.72 (4.93) | 1.26 (1.17) | −0.21 |
| 6. Everyone has assignments that are not completed on time or are behind schedule (reversed question) | 2.97 (3.8) | 1.3 (1.54) | −0.83 |
| 7. When participating in group-based learning activities, everyone can complete their work properly and efficiently. | 4.94 (5.11) | 1.33 (1.13) | −0.17 |
| 8. In the collaboration process, work is divided properly. | 5.03 (5.2) | 1.38 (1.11) | −0.17 |
| 9. When working with teammates, everyone knows what they are responsible for. | 5.06 (5.18) | 1.29 (1.14) | −0.12 |
| 10. Everyone focuses on participating in the group's learning activities and not doing anything else. | 4.75 (4.66) | 1.34 (1.26) | 0.09 |

The construct mean = 5.01 (5.10) without Items 2 and 6 (reversed questions); * The mean increase is the difference between the average of the Engineering Ethics class and the mean of the five classes. The average value of the column of the mean increase is –0.09, and the standard deviation is 0.12; ** The data in parentheses are for the five classes.

At the level of recognition of intra-group communication, the mean of Item 8 is highest, 5.78, as indicated in Table 5. In the process of collaboration, the Engineering Ethics students attached the greatest recognition to discussing matters without personal attacks. In the seven-point performance degree, Items 1, 8, and 9 all achieved a high-degree performance with 5.6 points or more, and the other items fell in a medium to a high-performance range of 4.2 to 5.6 points. In addition, for the Engineering Ethics class, compared with the mean of the five classes, the nine items scored four (Items 2, 4, 6, 9), with a degree of recognition lower than that of the five classes. The mean of the mean increase column is 0.06, and that of the standard deviation is 0.09. Compared with the five classes, the strengths of the research subject are one and eight, and the weak points are two, four, and six. In addition, the strengths and weaknesses of each item are used to contrast the intra-group communication of the classes. Item 1 shows the strength to consult a partner's opinions, and the weakness is two (understanding their opinions). Although students can provide real-time useful information (Item 3), this information may not be sufficient (weak point: Item 4) and the students may not be brave enough to express their opinions (weak point: Item 6). This result shows that students have made efforts in communication but have difficulty in understanding each other and even more difficulty in engaging in further demands. Finally, the construct's mean shows that the overall recognition of communication within the class (5.44) was higher than the mean of the five classes (5.38).

In terms of the degree of recognition of the Engineering Ethics class for intra-group trust, the mean of Item 4 was the highest at 5.92. This indicates that the research subjects are the most willing to help others through teamwork. Items 2, 1, and 5 follow next, with means higher than the construct mean of 5.65, as shown in Table 6.

**Table 5.** Data for the construct of intra-group communication.

| Items | Mean | Standard Deviation | Mean Increase * |
|---|---|---|---|
| 1. In the collaboration process, students will consult each other. | 5.69 (5.48) ** | 1.13 (1.09) | 0.21 |
| 2. In the collaboration process, students will understand each other's opinions. | 5.31 (5.34) | 1.27 (1.09) | −0.03 |
| 3. In the collaboration process, students will provide instant and useful information. | 5.53 (5.39) | 1.12 (1.13) | 0.14 |
| 4. In the collaboration process, students will provide sufficient information. | 5.22 (5.25) | 1.29 (1.06) | −0.03 |
| 5. In the collaboration process, all communication channels are sufficient and smooth. | 5.31 (5.26) | 1.05 (1.09) | 0.05 |
| 6. Students actively express their opinions. | 5.03 (5.06) | 1.14 (1.21) | −0.03 |
| 7. Students can accept different opinions. | 5.50 (5.38) | 1.17 (1.12) | 0.12 |
| 8. When confronted with controversial issues, students can discuss matters without personal attacks. | 5.78 (5.63) | 1.20 (1.10) | 0.15 |
| 9. When the students have different opinions, they can coordinate everyone to reach a consensus. | 5.61 (5.63) | 1.03 (1.00) | −0.01 |

The construct mean for intra-group communication is 5.44 (5.38). * The mean increase is the difference between the means of the Engineering Ethics class and the mean of the five classes. The average value of the column of the mean increase is 0.06, and the standard deviation is 0.09. ** The data in parentheses are for the five classes.

**Table 6.** Data for the construct of intra-group trust.

| Items | Mean | Standard Deviation | Mean Increase * |
|---|---|---|---|
| 1. When participating in group learning activities, I believe that other partners will do their best. | 5.75 (5.69) ** | 1.04 (1.08) | 0.06 |
| 2. When participating in group learning activities, I believe that we will collaborate successfully with each other. | 5.83 (5.65) | 1.01 (1.05) | 0.18 |
| 3. When teammates give their opinions, I will not question their motivations. | 5.28 (5.34) | 1.37 (1.25) | −0.06 |
| 4. When a teammate encounters a problem while studying, I will help him or her solve it. | 5.92 (5.74) | 1.06 (1.04) | 0.18 |
| 5. When I encounter problems in my studies, I will actively seek help from my teammates. | 5.69 (5.58) | 1.17 (1.19) | 0.11 |
| 6. I often feel that my teammates support or encourage each other. | 5.53 (5.49) | 1.09 (1.17) | 0.04 |
| 7. My teammates are very close. | 5.53 (5.53) | 1.19 (1.14) | 0 |
| 8. I can often feel the teacher's concern for my teammates. | 5.64 (5.59) | 1.08 (1.23) | 0.05 |

The construct mean of intra-group communication is 5.65 (5.58). * The mean increase is the difference between the mean of the Engineering Ethics class and the mean of the five classes. The mean of the column of the mean increase is 0.07, and the standard deviation is 0.08. ** The data in parentheses are for the five classes.

On the seven-point scale, Items 4, 2, 1, 5, and 8 all achieved a high performance of 5.6 or higher, and the other items fell in the medium- to high-performance range of 4.2 to 5.6. In addition, when comparing the Engineering Ethics class with the five classes, among the eight items, only Item 3 has a mean of recognition lower than the mean of the five classes. The mean of the column of the mean increase is 0.07, and the standard deviation is 0.08.

Compared with the means of the five classes, the strengths of the research subjects are Items 2, 4, and 5. The weak point is Item 3. Finally, the construct mean of intra-group trust (5.65) is slightly higher than the mean of the five classes (5.58), and the seven-point performance achieved a performance of 5.6 points or higher.

In general, for the constructs of coordination, communication, and trust, the means of intra-group trust, whether for the Engineering Ethics class (5.65) or among the five classes (5.58), are the highest among the three constructs. For the Engineering Ethics class, this value is slightly higher than that for the five classes. On the contrary, the means of intra-group coordination are lowest both in the Engineering Ethics class (5.01) and among the five classes (5.10). For the Engineering Ethics class, this value is even lower than that for the five classes. Finally, the means of intra-group communication are between those of communication and trust, and the value of the Engineering Ethics class (5.44) is slightly higher than that of the five other classes (5.38).

In terms of coordination, compared with the values of the five classes, the strengths of research subjects 1 (group meetings always start on time) and 10 (focus on participating in group learning activities) are higher than the average of the five classes. However, Items 5 (Interested in participating in group meetings), 7 (appropriate and efficient completion of work), and 8 (work is properly arranged) are significantly lower than the average of the five classes. As shown by the coordination, the research

subjects value discussions with focus and starting meetings on time, whereas interest in meetings, the efficiency of completing work, and work arrangement are relatively weak.

In the construct of intra-group communication, compared with the means of the five classes, the students made substantial efforts in communication (strong point on Item 1: consulting the other partners' opinions). However, they have difficulties in understanding each other (the weak point for Item 2: understanding the other partners' opinions), as well as difficulties in putting forward further needs (the weak point for Item 4: Insufficient information; the weak point for Item 6: be brave in expressing one's opinions).

In the construct of intra-group trust, compared with the means of the five classes, the class's strengths are Items 2 (I believe we will cooperate successfully), 4 (When a classmate encounters a problem, I will help him solve it), and 5 (I proactively seek help from my classmates), while the Item 3 is weak (I will not question the other partner's motivations). This implies that the class recognizes the success of collaboration and will actively assist and ask for help. However, they doubt each other's motives, relatively speaking.

### 3.2. Taiwan Energy Perception Survey

In order to understand the energy literacy of the research subjects, this study applied the "Taiwan Energy Transition Public Perception Survey" developed by the National Taiwan University Risk Society and Policy Research Center [32]. In June 2018, the center commissioned Chunghwa Telecom to conduct a Taiwanese energy policy perception survey using the "stratified random sampling method" for people over 18 years old in Taiwan via telephone interview. The sample number was 1068, and the sampling error was ±2.98%.

This research surveyed the subjects using the same questions via digital questionnaires on the internet. All quantifiable questions used a five-point scale based on the original questionnaire from the Risk Society and Policy Research Center. The quantitative description of the questionnaire is shown below. A total of 59 course students were used, and 48 questionnaires were collected. Among them, the question types that were most relevant to college students were selected for comparison in this paper. This paper will discuss three issues of energy perception: general questions, external cost internalization, and energy-saving life.

### 3.2.1. General Questions on Energy Perception

Table 7 provides a comparison of the perception of energy policy in Taiwan between the research subjects and the public. The means of Questions 1 and 2 are higher among the subjects than those of the public in Taiwan. The results show that the students are less concerned about the impacts of climate change on Taiwan and Taiwan's energy policy than the general public and understand the green energy policy of 2025 even more poorly (Question 3). Although these data cannot be compared with statistical significance, these results still illustrate the simple state of self-perception between the subjects and the general public in Taiwan.

In addition, if the value of each interval is 5/5 = 1 on a five-point scale, the degree of difference is as follows:

High-level performance: from 0 to less than 1;

Medium-to-high-level performance: from one to less than two;

Moderate-level performance: from two to under three;

Low- to medium-level performance: from three to less than four points;

Low-level performance: four or higher.

Inserting the levels of performance into the statistical results (Table 7), the research subjects are shown to have the same level of performance as the public in Taiwan for the first three questions. In other words, the impact of climate change on Taiwan is moderate to high, and the level of concern for Taiwan's energy policy is only moderate. The level of understanding of green energy policies in 2025 is

at a low level, and the fairness and planning of the attributes of Taiwan's current energy transition policies are moderate, while the urgency is medium to high.

In addition, for the question "I know the major energy resource in Taiwan", the correct answer is coal; the number of incorrect answers was 17, and the number of correct answers was 31 for the research subjects. The correct answers accounted for 65% of the 48 respondents. Compared with the 32% accuracy rate of the public in Taiwan measured by the risk center, the Engineering Ethics class scored much higher.

### 3.2.2. External Cost Internalization

In the questions related to "internal cost internalization (1)" in Table 8, "willing to pay the annual increase in electricity prices" mostly falls between "unwilling to pay too much" (33.3%) and "2.7 to 3.0 yuan" (31.3%). Compared with the public people in Taiwan, the subjects score relatively low in this area. The main reasons for a willingness to increase one's electricity prices are "protection of the environment" and an "increase of energy-saving incentives", which are consistent with the results for the public people in Taiwan.

In the "will pay higher electricity prices to support renewable energy" category, the majority of the class chose ordinary (37.5%), compared with the public in Taiwan, who chose willingness (45.5%). Again, this shows that the class is less willing to pay higher electricity prices than the public. In the question of "The range of oil price promotion because of energy tax", the class mostly chose 0.5 NTD (52.1%), which was also the highest for the public in Taiwan, but the ratio (37.6%) was lower than that for the class (Table 9).

### 3.2.3. Energy-Saving Life

For contributions to "Energy Conservation Actions" (Table 10), the research subjects present the highest ratio for "turn off lights and electrical appliances" (91.7%), which is also the highest for the public in Taiwan (68.5%). Among the "factors that can improve energy conservation", the class noted "products with clear and easy-to-understand energy-saving information" (68.8%) as the highest, while the nation's population chose "enjoy subsidies when purchasing energy-saving products" as the highest (55.2%).

The comparisons of energy perceptions between the class and the public in Taiwan are summarized in Table 11 under three categories: similar, better, and lower. Overall, most items are similar between these two groups. Both recognized that climate change has a moderately high impact on Taiwan, while showing only a moderate degree of concern for Taiwan's energy policy. Furthermore, both groups' understanding of the green energy policy for 2025 dropped to a low level. The reasons for the respondents' willingness to increase their electricity prices are both mainly "environmental protection" and "enhancement of energy-saving", and "the magnitude of the increase in the price of oil received by the promotion of energy taxes" is 0.5 NTD for the majority of both groups.

**Table 7.** Taiwanese Energy Policy Perception Survey.

| | 1. I Think about the Degree of the Impact of Climate Change on Taiwan | 2. I Care about Taiwan's Energy Policy | 3. My Understanding of the Green Energy Policy of 2025 | 4.1. Please Intuitively Evaluate Taiwan's Current Energy Transition Policy Attributes—Fairness | 4.2. Please Intuitively Evaluate Taiwan's Current Attributes for its Energy Transition Policy—Plannability | 4.3. Please Intuitively Evaluate the Current Attributes of Taiwan's Energy Transition Policy—Urgency |
|---|---|---|---|---|---|---|
| Quantitative description | 1 Very influential | 1 Very concerned | 1 Very clear | 1 Very fair | 1 Very planned | 1 Very urgent |
| | 5 Very uninfluential | 5 Very unconcerned | 5 Very unclear | 5 Very unfair | 5 Very unplanned | 5 Very unurgent |
| Mean | 1.54 | 2.54 | 3.17 | 2.00 | 2.00 | 1.44 |
| Performance Level | Medium High | Moderate | Medium Low | Medium | Medium | Medium High |
| Standard deviation | 0.54 | 0.65 | 0.93 | 0.58 | 0.85 | 0.97 |
| National average | 1.49 | 2.05 | 3.19 | * | * | * |

* Different scales and not compared.

**Table 8.** Comparison of "External Cost Internalization" questions (I).

| I Am Willing to Pay an Annual Increase in Electricity Prices | Number of People from the Research Subjects | Percentage of People from the Research Subjects (%) | Percentage of People in Taiwan (%) | Reasons I Am Willing to Increase My Electricity Prices (Choose Two) | Number of People from the Research Subjects | Percentage of People from the Research Subjects (%) | Percentage of People in Taiwan (%) |
|---|---|---|---|---|---|---|---|
| 2.7~3.0 NTD | 15 | 31.3 | 47.9 | protect the environment | 30 | 62.5 | 52.3 |
| 3.0~3.5 NTD | 6 | 12.5 | 16.5 | Increase incentives to save energy | 20 | 41.7 | 28.7 |
| 3.5~4.0 NTD | 0 | 0.0 | 6.1 | Reducing the risk of nuclear disasters | 10 | 20.8 | 26.8 |
| 4.0~4.5 NTD | 1 | 2.1 | 2.8 | Reduce high energy consumption industries | 15 | 31.3 | 14.6 |
| Above 4.5 NTD | 1 | 2.1 | 3.1 | other | 3 | 6.3 | 0.5 |
| unclear | 9 | 18.8 | 4.8 | Unwillingness to raise electricity prices | 12 | 25.0 | 20 |
| Unwilling to pay too much | 16 | 33.3 | 19.0 | | | | |
| Sum | 48 | 100 | 100 | | 48 | 100 | 100 |

**Table 9.** Comparison of "External Cost Internalization" questions (II).

| Willing to Pay Higher Electricity Prices to Support Renewable Energy | Number of People from the Research Subjects | Percentage of People from the Research Subjects (%) | Percentage of People in Taiwan (%) | The Range of Oil Price Promotions Due to an Energy Tax | Number of People from the Research Subjects | Percentage of People from the Research Subjects (%) | Percentage of People in Taiwan (%) |
|---|---|---|---|---|---|---|---|
| Very willing to | 4 | 8.3 | 15.2 | 0.5 NTD | 25 | 52.1 | 37.6 |
| Willing | 15 | 31.3 | 45.5 | 1.5 NTD | 5 | 10.4 | 15.5 |
| Ordinary | 18 | 37.5 | 1.2 | 2.0 NTD | 6 | 12.5 | 11.2 |
| Unwilling | 10 | 20.8 | 21.2 | 4 NTD | 2 | 4.2 | 4.7 |
| Very unwilling | 1 | 2.1 | 13.9 | Above 4 NTD | 0 | 0.0 | 5.5 |
| No opinion | 0 | | 3.0 | Others | 2 | 4.2 | 5.2 |
| | | | | Unwilling to pay oil price promotions because of the energy tax | 8 | 16.7 | 20.3 |
| | 48 | 100 | 100 | | 48 | 100 | 100 |

**Table 10.** Comparison of "Energy-Saving Life" questions.

| My Contribution to Energy Conservation Actions | Number of People from the Research Subjects | Percentage of People from the Research Subjects (%) | Percentage of People in Taiwan % | I Think the Factors That Can Improve Energy Conservation (Choose Two) | Number of People from the Research Subjects | Percentage of People from the Research Subjects (%) | Percentage of People in Taiwan (%) |
|---|---|---|---|---|---|---|---|
| Turn off unused lights and appliances | 44 | 91.7 | 68.5 | Enjoy subsidies when purchasing energy-saving products | 27 | 56.3 | 55.2 |
| Increase air-conditioning temperature + use fan | 21 | 43.8 | 45.2 | Products with clear and easy-to-understand energy savings information | 33 | 68.8 | 45.9 |
| Eliminate old appliances and replace them with energy-efficient appliances | 26 | 54.2 | 39.3 | Media announces how to save energy | 22 | 45.8 | 38.1 |
| Less air-conditioning | 24 | 50.0 | 36.6 | The government raising oil and electricity prices | 7 | 14.6 | 19.7 |
| Remind relatives and friends to turn off unnecessary appliances | 26 | 54.2 | 30.2 | Energy savings commissioner visits home to check the main reasons for power consumption | 16 | 33.3 | 17.1 |
| Take more public transportation No action | 18 | 37.5 | 28.4 | | | | |
| No action | 1 | 2.1 | 0.7 | | | | |
| SUM | 160 | 333.3 | 248.9 | | 105 | 218.8 | 176 |

**Table 11.** The comparison of the energy perceptions between the class and the public in Taiwan.

| The Class Performance | Items |
| --- | --- |
| Similar to the public in Taiwan | Consider the impact of climate change on Taiwan<br>Level of concern for Taiwan's energy policy<br>Understanding level of the nuclear-free energy policy of 2025<br>The reasons for their willingness to increase electricity prices are mainly "environment protection" and the "enhancement of energy savings"<br>The magnitude of the increase in oil prices as a result of energy tax incentives |
| Better than the public in Taiwan | Correct answer to "the current major energy resource in Taiwan" |
| Lower than the public in Taiwan | Willingness to pay for annual increases in electricity prices<br>Willingness to pay higher electricity prices to support renewable energy |

The class had a higher correct ratio for "the current major energy resource in Taiwan." However, their willingness to "increase annual electricity prices" and "pay higher electricity prices to support renewable energy "is lower than that of the public in Taiwan, showing that the students' energy awareness is better but that they have less support for higher payments.

*3.3. Some Issues during Research Execution*

3.3.1. Difficulties in Dual-Topic Progression

One of the research goals was to investigate the feasibility of applying competency-oriented content into an Engineering Ethics course and proposing suggestions for the implementation of Engineering Ethics courses in the future. During the research, one difficulty for the researchers was that the course had to cover two topics simultaneously. One topic related to competencies like collaboration while the other involved cognitive topics like Engineering Ethics case studies. Although the knowledge of collaboration was lectured upon at the beginning of the semester, and teamwork was demanded by group activities and subsequent assessments, due to other cognitive topics (such as Engineering Ethics case studies or the energy transformation justice of the algal reef), the course progress and requirements had to cover these two topics simultaneously (collaboration and cognitive topics). Could this dual-topic progression confuse students in their learning process? Meanwhile, the lecturer was also aware of the difficulty in manipulating two topics simultaneously.

The strategy for this research was to engage in collaborative lectures and workshops at the beginning of the semester and apply teamwork evaluation only later in the cognitive topic courses. It is crucial to organize the schedule and activities to properly realize the competency components and cognitive materials. In this study, collaboration competency was carried out for the whole semester in different forms, including lectures, workshops, class activities, and a final project evaluation. Energy literacy, on the other hand, was executed later than the collaboration competency via a case study, field visits, and a final project.

The final and most important suggestion for a future course is to emphasize the attitude of the lecturer, not just the students. It is not easy to handle dual topics simultaneously. In addition, collaboration competency must be developed gradually; therefore, the lecturer has to repeat the evaluation standard for teamwork to intensify the students' habits of team collaboration. Therefore, instead of focusing on students, it is more important to ensure that the lecturer is used to demanding teamwork in the classroom regularly for a competency-oriented course.

3.3.2. A High Percentage of Invalid Questionnaire Samples

Should reverse questions be applied to detect invalid samples? Indeed, reverse questions can be confusing for students, as they might mistakenly fill in the answers along with the previous question.

How, then, do we identify invalid samples if students answer questions randomly in the absence of reverse questions?

The author sent 244 questionnaire forms to third-year in 2019. In total, 1/4 of the questionnaire forms were invalid, including contradictions in the reverse questions, all the same choices in one construct, etc.

Some recommendations for this issue are as follows: (1) Reduce unnecessary questionnaires; (2) offer brief and sufficient explanations before each survey, including the relevant attitudes, score allocation methods, and the reasons that cause an invalid form; (3) design non-reverse debugging questions (for example, "This is a debugging question; please choose 1 to make this form valid"); (4) implement a scoring system, integrate surveys into class activities, and offer survey feedback afterwards. If students were to find the meanings of the survey applicable to their own lives, they would no longer find it meaningless to fill out the questionnaire and would be more willing to answer the questions. It is important to allow students to see the impacts of the questionnaire and develop responsible attitudes in their own lives.

## 4. Conclusions and Suggestions

Based on the collaboration and energy competency surveys of the research subjects, the following three sections illustrate our conclusions.

### 4.1. Comparison of Collaboration Competency between the Research Subjects and the General Mechanical Engineering Students

Overall, the collaboration questionnaire showed that among the three constructs, both the Engineering Ethics class (5.65) and the five classes had the highest mean for "intra-group trust" (5.58), followed by "intra-group communication", with 5.44 for the Engineering Ethics class, which is slightly higher than the mean of the five classes (5.38). Finally, "intra-group coordination" has the lowest results, with 5.01 for the mean of the Engineering Ethics classes and 5.10 for that of the five classes. This shows that the trends are similar among the three constructs of the collaboration questionnaire. Among them, "trust" reached a high level of performance, while "communication" and "coordination" fell between a medium and high level of performance, with 4.2 to less than 5.6 points.

Compared with the average of the five classes, for intra-group coordination, the research subjects recognized most strongly that they should start meetings on time and prepare well for their group meetings with a clear agenda. However, their interest in such meetings, their efficiency in completing their work, and their work schedules are relatively weak. In terms of communication, the classmates make substantial efforts (strong point: Item 1: consulting partner's opinions) but have difficulties in understanding each other (weak point: Item 2: understanding partner's opinions) and requesting further demands (weak point: Item 4: insufficient information; weak point: Item 6: be brave in expressing opinions). In terms of trust, the class recognized the success of collaboration most strongly and will actively assist and seek assistance but consciously doubt one another's motives.

Based on the results of the collaboration questionnaire, the construct of intra-group coordination is weakest and should be emphasized in future collaboration courses, especially regarding items like increasing interest in participating in group meetings and group learning activities and completing work properly and efficiently.

### 4.2. Comparison of Energy Competencies between the Research Subjects and the Public People in Taiwan

Based on the Taiwan energy perception survey, the perception of the research subjects is mostly similar to that of the people in Taiwan. For example, they both believe that the impact of climate change on Taiwan is at a medium to high level, but they are only moderately concerned about Taiwan's energy policy, while their understanding of the green energy policy for the year 2025 declines to a low to medium level. The reasons for the subjects' willingness to increase their electricity prices are mainly "environmental protection" and the "enhancement of energy-saving", while 0.5 NTD was

chosen mostly for "the magnitude of the increase in the price of oil received by the promotion of energy taxes" for the two groups.

The subjects' presented superior performance to the people in Taiwan in their ratio of correct answers for the question "What is the major energy resource in Taiwan?". However, their willingness to "increase electricity prices annually" and "pay higher electricity prices to support renewable energy" are lower than those of the national population, showing that the students' energy awareness is superior but with less support for higher electricity prices. However, these are not accurate survey results due to their finite numbers of questions. Nevertheless, these results suggest that the course students may better recognize important energy issues but lack the willingness to implement relevant changes. As a result, to facilitate the education of energy literacy (except to understand the nature and role of energy in the university and our lives), it is important to emphasize decision-making about energy use based on an understanding of energy's impacts.

### 4.3. Suggestions for Competency Implementations in a Future Course

This study applied collaboration and energy competency to one college course; the future amelioration of some course implementation problems (like the confusion of dual-topic progression) could help realize the multiple goals of the course simultaneously. One practical method applied in this research is to hold collaborative lectures and workshops at the beginning of the semester, and apply teamwork evaluations later into cognitive topic courses. These cognitive topics are all energy-based to better incorporate energy literacy. Therefore, the schedules of different topics should be organized properly. Most importantly, competency cannot be fostered in the short-term, as it must change a learner's lifelong behavior. Hence, before other habits are picked up by students, the lecturer should instill a habit to demand collaboration in the classroom regularly for competency development in the course.

The final issue relates to the highly invalid questionnaire forms. Some recommendations include: (1) reducing unnecessary questionnaires; (2) offering brief and sufficient explanations before each survey; (3) designing non-reverse debugging questions; (4) implementing a better scoring system, and linking the meanings of the survey to students' lives. Students should be able to see the impact of the questionnaire to develop responsible lifelong attitudes.

**Funding:** The authors would like to thank the financial support from the Ministry of Education (Republic of China, Taiwan), The Teacher Team Empowerment Project on an Issue-Oriented Approach to Narrative Competence Development, Mechanical Project Empowerment for Future Narrative Talent Development.

**Conflicts of Interest:** The authors declare no conflicts of interest.

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
