# Peer review of "A Pilot Study to Incorporate Collaboration and Energy Competency into an Engineering Ethics Course"

_education, doi:10.3390/educsci10030072_

Round 1
Reviewer 1 Report
I read through this paper and found it very difficult to read due to grammatical and English language issues or wording choices. This paper needs significant English editing to make it more readable in order to fully comments on the scientific aspects.
In the end, it still wasn't clear to me how this method and results can be translated to contexts outside of Taiwan. More explanation of the applicable lessons learned would be helpful to readers.
Intro:
First paragraph: The leap from ‘fake news’ to competencies is not well done. How does a competency framework lead students to understanding information and creating meaning?
Page 2, last paragraph: Citation for claim that education in Taiwan is different? Is there evidence that they do not do well in society?
Page 3, paragraph 1: “constructions” should be “constructs”. Need citation at end of sentence for constructs. Next sentence “it”, what “it”? the competency helps the students?
Page 3, paragraph 2: missing many citations to support claims. What is “non-single major”?
Page 3, last paragraph: poor word choice, “difficult”, citation after second sentence missing
Research Framework:
First paragraph: Not sure what is meant by “is easy to be dogma”?
Second paragraph: wording, “is like”
Research object: do not understand the use of object, do you mean subject?
2.1 paragraph p4, Wording, “males in terms of gender”
2.2 Research purpose: I don’t understand the purpose of this paragraph
Table 1. Oddly formatted, it’s hard to tell where the line breaks are versus a new row. What is “Walking on the field”?
2.4. Wording issues throughout, refer to Table 3, I think you mean table 2?
3.2 First paragraph: this seems more like intro for methods. How were each of the methods integrated into the tool?
3rd paragraph: repeated information but last time it was 93.1 % male, now 93.2 % male
Table 4 & 5: strange formatting for results, I don’t understand what the 5.33 means if the 5.23 is the class average across all??
3.3.2 When you say ratio of people, do you mean percentage of people?
Table 7: I do not understand the reporting of the results
Reviewer 2 Report
In my opinion the paper contains an interesting statistical analysis. However I think that it lacks fundamental elements, as I try to summarize below:
- The title does not seem to fully fit with the paper's contents.
- The abstract is not properly organized and contains some ideas that apparently should not be within an abstract.
It is my understating that it should be more condensed paper version.
I do not think that the paper’s purpose is clear.
3.My main comments to the introduction are as follows:
- There is no connection with the limitations of previous research, i.e. no reasoning to justify the need for the article to improve previous research.
- Main results and findings deriving from the research are not mentioned.
- I would propose adding a purpose statement as well as concrete research questions.
- Bibliographic mentions could be more rigorous.
- Point 2.2 could better explain the paper’s purpose.
- Description of results in 3.1 is not detailed and comprehensive and statements are not as in-depth as it would desirable.
Problem described in 3.4.1 under the heading “Confusion of dual-topics progression” does not facilitate the paper’s understanding .
A significant percentage of invalid questionnaire samples results in unsolved problems.
- Conclusions do not respond to goals followed by the study.
Moreover I must add that conclusions are not a synthesis of key research issues.
- It would be of interest that the paper suggests additional research to be undertaken. A more detailed bibliography section is missed.
Round 2
Reviewer 1 Report
The paper still lacks organization. For instance, the paper spends several paragraphs setting up competencies as a focus, which is much improved but then jumps to energy with no transition on page 3. There should be a section break and a transition to set up how energy is going to be framed as the domain to which the competency will be assessed.
Page 19 is where I think finally understood the purpose of the paper, "One of the research goals was to investigate the feasibility of applying competency-oriented contents into an Engineering Ethics course and proposing suggestions for the implementation of Engineering Ethics courses in the future." This needs to be emphasized in the introduction. However, this is in conflict with the energy focus previously. Is it ethics or energy? This needs to be clarified and used consistently in the paper.
You spent a paragraph in the intro talking about how the competency was needed so that you didn't say that they achieved a course goal when only satisfying one skill or topic, but it seems like your analysis wasn't consistent with that perspective.
There is an inconsistent font formatting in the paper, which was very distracting.
Author Response
Thanks for the commends of the reviewer. The spelling of the whole text has been checked again and some errors were corrected. And please check out the attachment and the edited paper to find the modification.

Reviewer 2 Report
In the updated version were added key findings and conclusions in the abstract.
References have been updated as requested by the evaluator.
The methodology is adequate and the results achieved are reliable with respect to the method used.
Author Response
Thanks for the commends of the reviewer. The spelling of the whole text has been checked again and some errors were corrected.